

# The role of European windstorm clustering for extreme seasonal losses as determined from a high resolution climate model

Matthew D. K. Priestley[1], Helen F. Dacre[1], Len C. Shaffrey[2], Kevin I. Hodges[1,2], and Joaquim G. Pinto[3]

[1]Department of Meteorology, University of Reading, Reading, UK
[2]NCAS, Department of Meteorology, University of Reading, Reading, UK
[3]Institute of Meteorology and Climate Research, Karlsruhe Institute of Technology, Karlsruhe, Germany

**Correspondence:** Matthew Priestley (m.d.k.priestley@pgr.reading.ac.uk)

**Abstract.**

Extratropical cyclones are the most damaging natural hazard to affect western Europe. Serial clustering occurs when many intense cyclones affect one area in a period of time which can potentially lead to very large seasonal losses. Previous studies have shown that intense cyclones may be more likely to cluster than less intense cyclones. We revisit this topic using a high
resolution climate model with the aim to determine how important clustering is for windstorm related losses.

The role of windstorm clustering is investigated using a quantifiable loss-based metric (storm severity index (SSI)) based on near-surface meteorological variables (10-metre wind speed) that is used to convert a wind footprint into losses for individual windstorms or seasons. 918 years of high resolution coupled climate model data from the High-Resolution Global
Environment Model (HiGEM) are compared to ERA-Interim re-analysis. HiGEM is able to successfully reproduce the wintertime North Atlantic/European circulation, and represent the large-scale circulation associated with the serial clustering of European windstorms. We use two measures to identify any changes in the contribution of clustering to the overall seasonal losses for increasing return periods.

Above a return period of 3 years, the accumulated seasonal loss from HiGEM is up to 20% larger than the accumulated seasonal loss from a set of random realisations of the HiGEM data. Seasonsal losses are increased by 10-20% relative to randomised seasonal losses at a return period of 200 years. The contribution of the single largest event in a season to the accumulated seasonal loss does not change with return period, generally ranging between∼0.25-0.5.

Given the realistic dynamical representation of cyclone clustering in HiGEM, and comparable statistics to ERA-Interim, we conclude that our estimation of clustering and its dependence on the return period will be useful for informing the development of risk models for European windstorms, particularly for longer return periods.



# 1 Introduction

Extratropical cyclones are the dominant weather hazard that affects western Europe. On average extrtropical cyclones cause ∼$1.6 billion of losses to the insurance industry per year (Schwierz et al., 2010) as a result of building damage and business interruption from severe wind gusts and large amounts of precipitation. The most severe individual storms can have much

greater impacts than what may be observed in an average year, for example storms Daria (25/01/1990), Kyrill (18/01/2007), and Lothar (26/12/1999) caused $5.1, $5.8, and $ 6.2 billion of insured losses respectively (Munich Re, 2015). The most severe seasons are often characterised by the recurrent influence of multiple cyclone events occurring in a short space of time, e.g. such as the winter of 2013/2014 (Matthews et al., 2014; Priestley et al., 2017b).

There have been several attempts to quantify losses associated with severe extratropical cyclones in re-analysis data and with data from General Circulation Models (GCMs) (e.g., Pinto et al., 2007; Leckebusch et al., 2007; Donat et al., 2011).These studies have primarily focussed on assessments of current climate loss potentials, and how these may alter under future climate conditions. These analyses commonly use loss proxies based on gridded meteorological data, such as the Storm Severity Index (SSI) (Klawa and Ulbrich, 2003). The SSI has been found to reproduce the inter-annual variability of windstorm losses in

Germany with a correlation of r=0.96 (Klawa and Ulbrich, 2003). This analysis can be performed on a seasonal basis (Pinto et al., 2007; Leckebusch et al., 2007), but also for individual events (Della-Marta et al., 2009; Karremann et al., 2014b, 2016).

North Atlantic winter cyclones have a tendency to occur in groups that affect specific regions within a given period of time. This process is known as serial clustering (Mailier et al., 2006). Serial clustering has been observed in re-analysis data sets in

multiple studies (Mailier et al., 2006; Pinto et al., 2013), and is a prominent feature of the cyclones that affect western Europe. Vitolo et al. (2009), Pinto et al. (2013), and Cusack (2016) provided evidence that the magnitude of serial clustering occurring over western and northwestern Europe may increase for more intense cyclones. Furthermore, a strong connection was found between the number of cyclones in a season and their intensities, particularly over the European sector (Hunter et al., 2016). Recent studies (Pinto et al., 2014; Priestley et al., 2017a) have been able to associate specific dynamical conditions with North

Atlantic cyclone clustering events. Periods of clustering are associated with a strong and straight North Atlantic jet stream flanked by anomalous amounts of Rossby wave breaking (RWB) on one or both sides of the jet. The amount of RWB on each side of the jet determines the angle of the jet and hence the location of clustering. More RWB to the south (north) drives the jet and storms further north (south), whereas RWB on both sides keeps the jet and storms constrained to a more central latitude. When these dynamical conditions persist for an extended period of time, it drives many cyclones towards the same location in

western Europe. Often these cyclones are members of a 'cyclone family', where cyclones form on the trailing cold fronts of mature cyclones further downstream (Bjerknes and Solberg, 1922). These events can have huge socio-economic impacts, for example the seasons of 1990, 1999 and 2013/14 were all characterised by this behaviour and resulted in insured losses of €20, €16, and €3.3 billion respectively, as well as numerous fatalities across Europe (Munich Re, 2015).



Despite previous studies assessing the return periods of European windstorm losses (Pinto et al., 2007; Leckebusch et al., 2007; Donat et al., 2011; Pinto et al., 2012) the importance of clustering to severe windstorm loss seasons across the whole of Europe has received less attention (notably Karremann et al., 2014b, a). In this study, we will further explore how clustering is associated with windstorm losses for Europe. With this aim, the historical reanalysis datasets provide a comprehensive and

are typically around 40-100 years in length. Due to the temporal limitations of reanalysis, accurate estimations of high return period storms (1 in 200 year events) are therefore not possible. Assessments of European wind storm losses for longer return periods using GCMs have been performed (Pinto et al., 2012; Karremann et al., 2014b, a). However, the aforementioned studies were mainly interested in investigating changes to windstorm losses under future climate conditions and all were performed with models with coarse horizontal resolution (ECHAM5/MPI-OM1, T63, ∼180 km in Europe (Roeckner et al., 2006)). In this

study the High-Resolution Global Environment Model (HiGEM, Shaffrey et al. (2009)) is used since it has higher horizontal resolution compared to the GCMs used in previous studies. In addition, particular focus is placed on evaluating HiGEM's ability to represent the behaviour of clustering as identified in Priestley et al. (2017a).

The main science questions that will be addressed in this study are as follows:

1. Is HiGEM able to capture the upper-tropospheric large-scale dynamics associated with European cyclone clustering?

2. Does the SSI calculated using HiGEM output provide comparable results for individual windstorms and seasonal accumulations, to those obtained from the ERA-Interim re-analysis?

3. Does windstorm clustering contribute more to seasonal losses for more severe European wintertime seasons?

The paper continues as follows. The data and methods used are described in section 2. The results follow in section 3, which starts with an evaluation of HiGEM, then an analysis of the SSI as a suitable metric for comparing windstorms in HiGEM and

ERA-Interim. Finally the importance of clustering is addressed for increasing return period winter seasons. The conclusions are presented in section 4.

## 2  Data & Methods

### 2.1  Datasets

The main data source for this work are simulations performed using HiGEM (Shaffrey et al., 2009), a fully coupled high resolu-

tion climate model based on the HadGEM1 configuration of the Met Office Unified Model (Johns et al., 2006). The horizontal resolution of the HiGEM atmospheric component is 0.83° latitude x 1.25° longitude (N144) (∼90 km in mid-latitudes) with 38 vertical levels up to 39 km. The horizontal resolution of the ocean component is $\frac{1}{3}^{\circ}$ x $\frac{1}{3}^{\circ}$ (∼30 km) and is considered to be eddy-permitting. A total of 918 years of HiGEM data are available with a 6-hourly temporal resolution. This data comes from a series of 4-member ensemble decadal hindcasts initialised between 1960 and 2006, and also four 59 year transient

experiments initialised in 1957. Full details of the data used are described in Shaffrey et al. (2017). HiGEM has been shown to have a good representation of the North Atlantic storm tracks, and also in the representation and distribution of extratropical





cyclones (Catto et al., 2010, 2011).

For comparison, the re-analysis from the European Centre for Medium Range Weather Forecasts (ECMWF) ERA-Interim dataset (Dee et al., 2011) is used. ERA-Interim data is available at a 6-hourly resolution, starting from January 1979 and has
a T255 spectral horizontal resoultion (∼80 km) with 60 vertical eta levels up to 0.1 hPa. Therefore, the resolution of ERA-Interim is comparable to that of HiGEM. In total 36 years of ERA-Interim data are used, from 1979 to 2015. As the focus of this study is on wintertime losses resulting from extratropical cyclones our analysis will be constrained to the months of December, January, and February (DJF).

## 2.2   Cyclone identification and tracking

To identify extratropical cyclones in both datasets we use the tracking algorithm of Hodges (1994, 1995) applied in the same way as Hoskins and Hodges (2002). This method tracks features using maxima in the 850 hPa relative vorticity field in the Northern Hemisphere. Prior to tracking the vorticity field is spectrally truncated to T42, which reduces the noise in the vorticity
field. The large-scale background is also removed by removing total wavenumbers $\leq 5$ in the spectral representation (Hoskins and Hodges, 2002). Maxima in the vorticity field (i.e. cyclonic features) are identified every 6 hours and formed into tracks. This is initially done by using a nearest neighbour approach to initialise the tracks, which are then refined by minimizing a cost function for track smoothness that is subject to adaptive constraints on the smoothness and displacement in a time step (Hodges, 1999). Pressure minima are also associated with the tracks using a minimization technique (Bengtsson et al., 2009)
within a 5° radial cap. In order to exclude very small scale, noisy tracks, only tracks that travel at least 500 km and have a lifetime greater than 24 hours are retained. In previous dynamical clustering studies (Pinto et al., 2014; Priestley et al., 2017a) the method of Murray and Simmonds (1991) has been applied. Both cyclone tracking methods perform similarly for the tracking and clustering of North Atlantic cyclones, though the Hodges (1994) method generally tends to produce lower clustering values over the North Atlantic/European sector (Pinto et al., 2016).

We follow the method of Priestley et al. (2017a) to calculate composite fields of RWB and the upper-level jet on clustered days. RWB is calculated using the 2-D Blocking Index method from Masato et al. (2013), which identifies overturning of potential temperature countours on the 2 PVU surface (dynamical tropopause; $1\ \mathrm{PVU} = 1 \times 10^{-6}\ \mathrm{K\ m^2\ kg^{-1}\ s^{-1}}$). The upper-level jet is identified as regions of high wind speed on the 250 hPa surface. These fields are composited for cyclones passing
through three 700 km radii at different latitudes centred on 5°W; 45°N, 55°N, and 65°N, to focus on the impact for various locations in western Europe.





## 2.3 SSI metric

The metric developed by Klawa and Ulbrich (2003) is used as a loss proxy for European windstorms. The SSI has been used in numerous other studies for similar purposes (Leckebusch et al., 2007; Pinto et al., 2007, 2012; Karremann et al., 2014b, a). It uses 10-metre wind speeds in its calculation of storm severity. We follow the approach of the population weighted SSI as used by Pinto et al. (2012) and Karremann et al. (2014b). The formulation of the SSI is defined in equation 1 and is constructed as follows:

- Losses due to wind occur on approximately 2% of all days (Palutikof and Skellern, 1991), therefore, for any losses to be produced the wind speed ($V_{i,j}$) must exceed the 98th percentile of the wind speed distribution.

- Buildings are generally constructed in such a way that they can sustain gusts that are expected locally. Hence, the 98th percentile is the local value ($V_{i,j}^{98}$). Following the method of Karremann (2015), if the 98th percentile is less than 9 m s$^{-1}$, the 98th percentile value is fixed at 9 m s$^{-1}$.

- Losses do not occur if the wind speed does not exceed the local threshold ($I_{i,j}$).

- The value of $\frac{V_{i,j}}{V_{i,j}^{98}}$ is cubed as this is proportional to the kinetic energy flux (Palutikof and Skellern, 1991; Lamb, 1991) and this introduces a realistic, strongly non-linear wind-loss relationship (Klawa and Ulbrich, 2003).

- Winds exceeding the 98th percentile that do not occur over land are ignored as they will not contribute to losses ($L_{i,j}$).

- Insured losses from windstorms are dependent on the location of insured property, which are proportional to the local population density. The SSI is scaled by the 2015 global population density at the corresponding grid-box ($pop_{i,j}$, (Center for International Earth Science Information Network - CIESIN - Columbia University, 2017)).

$$SSI = \sum_{i=1}^{N_i} \sum_{j=1}^{N_j} \left( \frac{V_{i,j}}{V_{i,j}^{98}} - 1 \right)^3 \cdot I_{i,j} \cdot L_{i,j} \cdot pop_{i,j} \tag{1}$$

$$V_{i,j} = \begin{cases} max\,(V_{i,j,t}) & \text{if } V_{i,j} \geq 9\,ms^{-1} \\ 9\,ms^{-1} & \text{if } V_{i,j} < 9\,ms^{-1} \end{cases}, \, t = \text{time period}$$

$$I_{i,j} = \begin{cases} 0 \text{ if } V_{i,j} < V_{i,j}^{98} \\ 1 \text{ if } V_{i,j} \geq V_{i,j}^{98} \end{cases}$$

$$L_{i,j} = \begin{cases} 0 \text{ over seas} \\ 1 \text{ over land} \end{cases}$$

$pop_{i,j} = \text{Population density per grid-box}$

Following the method of Karremann (2015) we have applied a $V_{i,j}^{98}$ of 9 m s$^{-1}$ to regions where the $V_{i,j}^{98}$ value is less than 9 m s$^{-1}$. Changing the value of $V_{i,j}^{98}$ provides a sensible threshold for regions where the actual $V_{i,j}^{98}$ is not a realistic threshold for





the onset of damage, such as Southern Europe and Iberia.

The SSI is calculated at every land grid point and is used in two different forms for the main analysis in this study. Following insurance industry naming conventions, the first approach will be to calculate the maximum loss event in a year (herein

referred to as the Occurrence Exceedance Probability (OEP)), and the second will calculate the total loss for an entire DJF season (herein referred to as the Annual Exceedance Probability (AEP)).

The OEP is calculated as the spatial sum of the maximum SSi within a 72-hour period (i.e. the maximum SSI calculated from the 6-hourly windspeeds in a 72-hour period, per grid point). The 72-hour time window is consistent with that used by

re-insurance companies for defining a particular event (Mitchell-Wallace et al., 2017). The region used to calculate the OEP is an adapted version of the Meteorologoical Index (MI) box applied by Pinto et al. (2012). Our OEP region extends from $10°W - 20°E, 35°N - 60°N$ and covers all of western Europe and most of central Europe. Our region differs from that of Pinto et al. (2012) as it extends further south to $35°N$, in order to encompass the Iberian peninsula and also all of Italy (shown by the black box in figure 4b).

The AEP is calculated in the same way as the OEP, except that instead of using the single 72-hour maximum wind footprint, it sums all the individual 72-hour maximum wind speed footprints in the 90-day winter period. In the calculation of the AEP all events are retained. The sensitivity to retaining all events is tested later.

## 2.4   Clustering measures

There are several ways to assess the clustering of windstorms, which give different information and perspectives. Described below are the three methods which will be used in this study.

### 2.4.1   Dispersion statistic

The first measure is the dispersion statistic ($\psi$). This is a measure of the regularity of cyclone passages at a particular gridpoint (equation 2) (Mailier et al., 2006). This relates the variance ($\sigma^2$) in storm track numbers to the mean ($\mu$), with positive (negative) values indicating that cyclones are more likely to occur in groups (regularly). Near-zero values indicate a more random occurrence of cyclones (corresponding to a Poisson distribution).

$\psi = \left(\dfrac{\sigma^2}{\mu}\right) - 1$             (2)



This statistical measure is the base quantification of where the dynamical clustering of cyclones is occurring. When done on a grid-point by grid-point basis it illustrates where cyclone passages are more regular, or more clustered and has been applied in numerous studies for this purpose (Mailier et al., 2006; Vitolo et al., 2009; Pinto et al., 2013). Moreover, Karremann et al. (2014b, a) estimated clustering of European storm series of different intensities and frequencies by approximating the data with
a negative binomial distribution, thus estimating the deviation from a random Poisson distribution.

### 2.4.2  AEP/AEP_random

Another measure for assessing the impact of the clustering of cyclones is to examine the ratio of the AEP to a randomised
version of itself (this will herein be referred to as AEP_random). The randomisation re-orders all of the 72-hour SSI periods in the 918 DJF periods from HiGEM. Artificial DJF seasons are constructed by randomly sampling thirty 72-hour periods into a new order to remove any clustering between events that may be present in the HiGEM climate model. The AEP/AEP_random measure of clustering is particularly important for re-insurers as it provides information on how having dynamically consistent years (e.g. from the HiGEM model) provides different AEPs relative to a set of random (stochastic) model year.

A positive value of AEP/AEP_random suggests that the dynamically consistent clustering and the severity of cyclones in HiGEM result in a larger AEP, relative to that expected from a randomly sampled set of events. Similarly, a negative value suggests that the consistent grouping of cyclones gives a lower AEP than would be expected at that particular return period.

### 2.4.3  OEP/AEP

The final measure used to assess clustering is the ratio of the OEP to the AEP. If the total loss in a season were characterised by just one single 72-hour cyclone event then, by definition, the AEP and OEP would be identical. However, if the OEP were much smaller than the AEP then this would suggest there are many cyclone events contributing to the AEP. The OEP/AEP ratio therefore quantifies the dominance of a single loss event in a season. The OEP/AEP ratio is calculated using the OEP and
AEP in the same season.

It should be noted that all the above measures provide different interpretations of the occurrence of clustering. The dispersion statistic is a measure of how grouped storms are in time relative to a Poisson distribution. This can be so physically interpreted as measuring the seriality of clustering. The ratio of AEP to AEP_random provides information on the dynamically
consistent grouping of cyclones affects the accumulated seasonal losses (e.g. that produced from a climate model) compared to a completely random series of cyclones. The OEP to AEP ratio gives information on the dominance of the largest loss event in the overall seasonal losses. One of the additional objectives of this study is to ascertain how consistent the different measures of clustering are for seasonal losses.





## 2.5 Return periods and statistical methods

A majority of the results in this paper will be expressed in terms of return period. The return period provides a period of time in which an event of a certain magnitude is expected to occur. Return periods have been allocated in a way such that the maximum AEP year is assigned a return period of the length of the dataset divided by its rank (for the maximum event the rank is 1). Therefore the maximum AEP year from ERA-Interim has a return period of 36 years, the highest AEP year in HiGEM has a return period of 918 years, as they both occur once in their total time period respectively. The second largest events then have return periods of 18 years and 459 years for ERA-Interim and HiGEM respectively. This continues until the lowest ranked year, which has a return period of 1 year. For a majority of our analysis we rank the OEP in the order of descending AEP. This ensures we maintain a temporal connection between the OEP and AEP at all return periods and means that the largest OEP may not necessarily occur in the highest AEP year. Some analysis is performed on independently ordered AEP and OEP, which removes the connection between maximum events and the years in which they occur.

The return periods of the most extreme events are estimated using a generalized Pareto distribution (GPD) that is fitted to the AEP and OEP data above a specified threshold ('peak over threshold' method). The GPD is fit using the maximum-likelihood method, following M. and G. (2009), and Pinto et al. (2012). Uncertainties at the 95% level are calculated using the delta method (Coles, 2001).

## 3 Results

## 3.1 Evaluation of cyclone clustering in HiGEM

HiGEM has a good representation of the large-scale tropospheric circulation (Shaffrey et al., 2009; Woollings, 2010) and also extratropical cyclone shape, structure, and distribution (Catto et al., 2010, 2011). Several other studies have demonstrated that the model has skill in seasonal to decadal predictability, particularly in the North Atlantic (Shaffrey et al., 2017; Robson et al., 2017). HiGEM provides a good representation of both the structure and amplitude of the DJF North Atlantic storm track (figure 1a, c, e). The characteristic tilt of the North Atlantic storm track is evident as cyclones in HiGEM (figure1c) follow the SW-NE path found in ERA-Interim (figure1a). In addition the maxima in storm numbers off the coast of Newfoundland and also over the Irminger Sea contrast well with ERA-Interim (figure1e). There is an anomalous extension of the storm track in its exit region in HiGEM across Denmark and northern Germany, however the amplitude of the anomaly is small (< 2 cyclones per month). On the larger scale there are minimal biases present across the entire basin and the European continent. There are localised errors that are mostly below 2 cyclones per month when compared to 36 years of ERA-Interim re-analysis (consistent with Catto et al. (2011)). The structure and amplitude of the Mediterranean storm track is also well captured. Stippling



in figure 1e indicates where HiGEM and ERA-Interim are different at the 95% level (performed using a two-tailed Student's t-test). These differences are only present around the coast of Greenland and are associated with the minima in the Labrador Sea and Davis Strait, and also the maxima across the east coast of Greenland. None of the anomalies across the rest of the North Atlantic or Europe are statistically significant.

The dispersion of cyclones in the North Atlantic for ERA-Interim and HiGEM is shown in figure 1b and 1d respectively. The pattern of the dispersion is consistent for the two datasets with both being characterised by a more regular behaviour in the entrance of the storm track (western North Atlantic) where storms have their main genesis region (Hoskins and Hodges, 2002) and baroclinic processes are dominant. Both datasets show overdispersive (clustered) behaviour in the exit of the storm track, e.g. the UK and Iceland. The pattern of under/overdispersion in the exit/entrance region of the storm track is comparable with the studies of Mailier et al. (2006); Pinto et al. (2013). There are discrepancies in the magnitude of the dispersion (figure 1f), however, the large-scale pattern and sign of the dispersion is consistent between the two datasets.

It was shown in Priestley et al. (2017a) that clustering events occurring at different latitudes of western Europe were associated with a strong and extended upper-level jet that was associated with anomalous RWB on one or both flanks, which acted to drive the jet further north or further south and then anchor it in position. These persistent conditions allow the cyclones to track in similar directions and leads to clustering over different regions of western Europe. This analysis has been repeated for all 918 years of HiGEM data (figure 2), with the same analysis of ERA-Interim being shown in the supplementary material (figure A1). Figure 2a shows that for cyclones clustering at 65°N, the jet is extended toward the northern United Kingdom with speeds in excess of 40 m s$^{-1}$. The jet is associated with large amounts of RWB on its southern flank. For clustered days at 55°N (figure 2b), the extended jet is more zonal than for events at 65°N. The strong jet has anomalous RWB on the northern and southern flanks, however the RWB on the southern flank is of a smaller magnitude than that seen in ERA-Interim (figure A1b). The clustered events at 45°N show a very zonal upper-level jet with a dominance of RWB on the northern flank.

All of the composites of clustered days (figure 2a-c) show a marked departure from the climatology (figure 2d). The jet is stronger and more zonally extended than the climatology for all three cases, and each features anomalous amounts of RWB on one or both flanks of the upper-level jet, all of which are comparable to ERA-Interim. The climatological state of the jet and RWB in HiGEM (figure 2d) is comparable with ERA-Interim (figure A1d), with a slightly reduced amount of RWB across central Europe. Hence, clustering events in HiGEM appear dynamically consistent with those in ERA-Interim, albeit with a slight under-representation of the anticyclonic RWB on the southern flank of the jet.

HiGEM has been found to have a good representation of North Atlantic extratropical cyclones and cyclone clustering, as well as the large-scale circulation driving this behaviour. This demonstrates the suitability of using HiGEM to investigate clustered windstorm related losses.



## 3.2 Comparison of SSI in ERA-Interim and HiGEM

The SSI is widely used for quantifying losses related to windstorms. We now compare the SSI for both HiGEM and ERA-Interim. The characteristic of the SSI is that it is calculated above a set threshold, the 98th percentile of the local distribution of 10-metre wind speed ($V_{i,j}^{98}$). The structure of $V_{i,j}^{98}$ for ERA-Interim, HiGEM, and the difference between the two datasets

is shown in figure 3. Both datasets show a similar large scale structure with maxima over the North Atlantic ocean and minima over the high orography of the Alps and Pyrenees. However, HiGEM values are systematically lower than ERA-Interim across almost all of Europe by 1-3 m s$^{-1}$ (figure 3c), whereas across the North Atlantic ocean the bias is smaller and slightly positive. This systematic difference suggests there may be differences in the boundary layer scheme of HiGEM compared to ERA-Interim as this bias is not present for wind speed at 850 hPa or higher (not shown). Similar differences have also been

found for 10-metre winds when contrasted with 925 hPa winds between four reanalysis datasets (Hodges et al., 2011).

To address the lower European windspeeds, a simple bias correction is applied to the 10-metre wind speeds in HiGEM. This is done by correcting the 10-metre wind speeds by the spatially averaged offset in the $V_{i,j}^{98}$ field for all land grid points within our area of interest (black box in figure 4b). As a result all HiGEM wind speeds are increased by 18.75% over land. The resulting

bias-corrected HiGEM 10-metre wind speeds will be called HiGEM_bc herein and its formulation is shown in equation 3. The corrected $V_{i,j}^{98}$ wind field (difference relative to ERA-Interim) is shown in figure 3d and shows much reduced differences across our core European region. In some regions (northern Germany, Benelux, northern and northwestern France) the differences have changed sign and are now positive, however this is balanced by the slightly negative regions, resulting in an overall neutral anomaly compared to ERA-Interim.

$$HiGEM\_bc = HiGEM * \frac{\overline{p_{98}^{ERA-i}}}{p_{98}^{HiGEM}} \tag{3}$$

Spatial maps of the DJF average SSI are shown in figure 4 and are consistent between ERA-Interim and HiGEM across large parts of northern and northwestern Europe. Both datasets show a peak in SSI across northwestern Europe from London across to northern and northwestern Germany, as would be expected from the population weighting. Other densely populated regions are also identifiable. There are further regions of noticeable SSI across Germany and extending west toward Russia. There are

also peaks in SSI across the Iberian peninsula and Italy.

Figure 5a shows how the AEP from ERA-Interim and HiGEM as a function of return period, up to a maximum return period of 36 years. Also shown in figure 5a are 10,000 bootstrap samples of the 918 years of HiGEM_bc AEP data in 36 year samples, and the associated 95% confidence intervals. The AEP from ERA-Interim is within the confidence intervals of the

HiGEM_bc samples at all return periods, with ERA-Interim being at the upper-end of the spread at the lowest return periods of 1-2 years, and in the middle of the spread for the remaining return periods. Figure 5a suggests that HiGEM_bc can capture the variation of AEP as a function of return period that is found in ERA-Interim.





In figure 5b we have ordered the AEP and OEP independently in order to assess how the SSI from ERA-Interim and HiGEM directly compare in magnitude at varying return periods. Figure 5b shows how the AEP and OEP change with return period for ERA-Interim, with the dots representing the raw data. Also shown in figure 5b are the ERA-Interim GPD fits of the ERA-Interim AEP and OEP, and associated confidence intervals (non-filled). Confidence intervals are only displayed above the threshold used to fit the GPD. The GPD provides a good estimation of the data above a 5 year return period, however due to the small amount of data used to fit the distribution, uncertainties start to become very large above a return period of 20 years. By a return period of 50 years they have diverged greatly.

Also shown in figure 5b are the 95% confidence intervals for the GPD fits (shaded regions) of the HiGEM_bc AEP and OEP (ordered independently). These are shown above a return period of 10 years. The confidence intervals for HiGEM_bc are much narrower than ERA-Interim, which is to be expected as there is considerably more data being used in the GPD fit (consistent with Karremann et al. (2014b)). For return periods for which both datasets have data (< 36 years) the GPD fit of the HiGEM_bc AEP and OEP are within the confidence intervals of the ERA-Interim AEP and OEP. This is also the case for return periods greater than 36 years for the AEP and the OEP, although the confidence intervals for the GPD fit for ERA-Interim become extremely large for return periods greater than 50 years. HiGEM_bc is therefore consistent with the accumulated seasonal losses and the individual events found in ERA-Interim. This suggests HiGEM is a useful climate model for investigating AEP and OEP for large return periods.

### 3.3 Large return Periods losses in HiGEM_bc

Figure 6 shows the AEP and OEP for HiGEM_bc. Both the AEP and OEP curves of HiGEM_bc extend beyond the respective maxima from ERA-Interim, suggesting that more severe windstorm seasons and also single cyclone events may be possible than those seen in the ERA-Interim period. For both the AEP and OEP, the increase in the largest event is by approximately 100%. There is more noise in the ERA-Interim curves compared to their HiGEM_bc counterparts due to the smaller number of years. The OEP and AEP of HiGEM_bc are sorted by AEP magnitude in figure 6. Consequently, there is substantially more spread in the OEP values than seen in figure 5b. However, a general increase in OEP with return period is still found, with low AEP years generally having a lower OEP and high AEP years having a higher OEP, but there are some specific deviations from this (for example, note the four very high OEP years between return periods 40 and 100).

To test the sensitivity of figure 5b and figure 6 to the default definition of AEP as the sum of all events, we repeated the analysis, but only retaining on average the top 3 events of each year. This equates to an order of magnitude reduction in the number of events (see figure A2). The magnitudes of the AEP are very marginally lower as would be expected with the filtering of events, but the main features of the curves in figure A2 are very similar.





(Hunter et al., 2016) previously showed how the number of cyclones within a winter is strongly related to their intensities. Similarly, earlier results (Fig. 1e) showed that cyclones in HiGEM tend to occur in groups. To quantify the contribution of windstorm clustering the large AEP values we compare the HiGEM AEPs to randomised series of loss events. The ratio be-
tween the AEP and the AEP_random may be particularly important for the insurance industry as it characterises the importance of the clustering of cyclones in seasonal losses.

We have performed a non-replacement randomisation of the 72 hour chunks that make up the HiGEM_bc AEP with 10,000 samples and this randomisation ensures that each random sample contains the exact same data as the original 918 years. This
randomisation allows us to assess how the intensity of losses and the associated number of cyclones acts to influence the AEP in HiGEM_bc, compared to a timeseries where windstorms are occurring randomly. The mean of these random samples is shown by the black line in figure 6, with the grey shading indicating the 95% confidence interval of these samples. Below a return period of ∼3 years the AEP_random is greater than that from the HiGEM_bc AEP and above the 3 year return period the AEP_random is consistently less than the AEP. Therefore, low (high) return period loss years tend to have a lower (higher)
AEP than random.

The ratio of AEP to AEP_random is shown in figure 7. Values >1 (<1) indicate a higher (lower) AEP in the actual HiGEM_bc years compared to the AEP_random years. Above a return period of 3 years the realistic representation of the grouping of events in HiGEM_bc tends to generate more losses and a greater AEP than the random grouping of events. The median contribution
above the 3 year return period is generally in the range of 1.1 to 1.2 times the random AEP. At a return period of 200 years the 95% confidence intervals range from 1 to 1.3. For low return periods (< 3 years) the occurrence of events in HiGEM_bc leads to a lower AEP than in the random realisations, with HiGEM_bc AEP values in the range of 0.6 to 1. This suggests that during low loss winters, there are a smaller number of windstorms and weather loss events occurring in HiGEM_bc than would be expected from considering a randomised series of events.

The difference can be interpreted physically by considering two recent DJF periods in the UK. Firstly, the winter of 2009/2010 was characterized by a strongly negative NAO and an absence of extratropical cyclones influencing the UK for this period (Osborn, 2011). Secondly, the winter of 2013/2014 was almost the complete opposite and was associated with the continuous presence of deep cyclones, occurring in groups, for almost the enter DJF period (Matthews et al., 2014; Priestley et al., 2017b).
The nature of these two seasons would result in 2009/2010 having a very low AEP, and 2013/2014 having a very high AEP. Randomising these two seasons the result would be two synthetic seasons with AEP values between these two extremes. Hence the clustering of cyclones in 2013/2014 results in a higher AEP than expected from random and 2009/2010 having a lower AEP than expected.

In figure 7 it appears that the realistic representation of clustering in HiGEM_bc causes higher losses above a 3 year re-





turn period with losses being 10-20% higher than random as a result at all return periods. Despite the nearly constant value of AEP/AEP_random above 3 year return period, the absolute difference between the two values is increasing with return period (the AEP is more than 4 times larger at 918 year return period compared to a 3 year return period). Hence the physically consistent representation of clustering in HiGEM_bc is causing larger increases to the AEP with increasing return period.

A different view of clustering can be gained by examining how a single event can affect the accumulated seasonal losses through the ratio of the OEP to AEP. A high value implies that the single largest event is causing most of the losses in a season, and a lower value implies a contribution to the overall seasonal losses from many cyclones in that particular season. The results from ERA-Interim and HiGEM are compared in figure 8a. This shows 10,000 random samples of 36 years of the HiGEM_bc OEP/AEP with associated confidence intervals as well as the ERA-Interim OEP/AEP. The values are sorted into return periods by order of descending AEP. There is considerable spread in the ERA-Interim OEP/AEP values, with minima of $\sim 0.2$ at a return period of 1-2 years, and a maxima of $\sim0.8$ at a return period of 10 years. There is no clear systematic increase or decrease in the value of OEP/AEP in the ERA-Interim data, which is consistent with the median values and confidence intervals from the HiGEM_bc samples. The 95% confidence intervals range from 0.15-0.65 at a return period of 1 year, and from 0.2-0.8 at a return period of 36 years. This suggests that there is a wide range of OEP/AEP values that may characterise a high or a low AEP season.

Figure 8b shows the OEP/AEP values for HiGEM_bc. As in figure 8a there is considerable spread variation in the value of OEP/AEP at all return periods, with no clear systematic increase or decrease with return period. All return periods have a majority of the data with values of $\sim0.25$-0.5, with the extremes ranging from 0.15 to 0.9. This indicates that in terms of the contribution of a single event to the overall seasonal loss, there is no direct relationship with return period. At any return period it appears that 25-50% of losses will come from the largest event.

As with the ratio of AEP/AEP_random in figure 7, the ratio of OEP/AEP in figure 8b has a relatively constant value at all return periods and this suggests a constant relationship between the OEP and AEP. However, as the return period is increasing the OEP and AEP are also increasing, so despite the AEP/AEP_random ratio being consistent at a return period of 5 years and 200 years, the absolute difference between OEP and AEP at the two return periods would be very different. The higher return periods have a higher absolute difference between the AEP and OEP, hence the additional losses that are not the OEP are increasing with return period. Hence the relative difference between the OEP and AEP is consistent with return period, but the absolute difference is continuing to increase with return period. This absolute increase is likely a result of more severe events in the high return period AEP years.



## 4    Discussion & Conclusions

The aim of this study is to investigate the importance of clustering for high return period loss events caused by European windstorms using a GCM that is able to adequately capture the large-scale dynamics controlling cyclone clustering. This work has been performed using HiGEM, a high resolution fully coupled climate model. The performance of HiGEM has been

evaluated using the ERA-Interim reanalysis. Losses from European windstorms have been estimated using a version of the SSI (Storm Severity Index) applied to European land grid points. The main conclusions of this work are as follows:

- HiGEM can successfully reproduce the large-scale dynamics associated with clustering of European cyclones that are seen in ERA-Interim. The biases in DJF storm track activity in HiGEM are small, with the tilt and intensity of the North Atlantic storm track being well represented. The pattern of dispersion in the North Atlantic is also consistent with ERA-

Interim, with cyclones clustering more near the exit of the storm track, and an underdispersive and regular nature in the entrance region. The large scale circulation associated with clustering is also similar in HiGEM and ERA-Interim. Both show how clustering in different locations of western Europe is associated with a strong and extended upper-level jet that is flanked on one or both sides by anomalous RWB. Hence, extratropical cyclone clustering in HiGEM is occurring for the right dynamical reasons.

- SSI is used as a proxy to assess losses occurring from intense European windstorms. The SSI is applied to land points only and for an area than encompasses all of western and most of central Europe. It is found that HiGEM systematically underestimates 10-metre wind speed over European land regions. A simple bias correction (scaling by 18.75%) leads to a structure of the DJF SSI average that is consistent between the bias corrected HiGEM (HiGEM_bc) and ERA-Interim. The return periods of AEP and OEP are found to be consistent between HiGEM_bc and ERA-Interim for return periods

less that 36 years. Therefore, HiGEM_bc appears to be a suitable model for assessing long return period losses from European windstorms.

- Compared to a random season of cyclones, the AEP from HiGEM_bc is larger at return periods greater than 3 years. The dynamically consistent representation of cyclone severity and clustering in HiGEM_bc results in values of AEP that are approximately 10-20% larger than AEP_random at a return period of 200 years. Therefore, not having a dynamically

consistent representation of cyclone clustering appears to result in losses be underestimated above a 3 year return period.

- The relative portion of the AEP that comes from the OEP is very variable across all return periods and there is no strong relationship between the two values. The contribution of the OEP to the AEP is found to be approximately ∼0.25 to ∼0.5 in HiGEM_bc. The absolute difference between the AEP and OEP is increasing with return period. Therefore, the relative influence of the largest loss event in a season does not change with return period.

In this study we have shown that having a dynamically consistent representation of cyclone clustering and storm intensity causes the AEP to be approximately 10-20% higher (for return periods greater than three years) than that expected from a random selection of cyclones. Despite the near constant values of AEP/AEP_random above a 3 year return period, the absolute



magnitude of the AEP relative to AEP_random is increasing with return period. This absolute increase suggests an increase in cyclone severity for higher return period loss seasons for cyclones of all magnitudes. This result has strong implications for loss modelling in the insurance industry and demonstrates that if a model does not adequately represent the clustering behaviour of cyclones then losses will be underestimated for larger return periods. In addition we have shown how the relative contribution

5  of the largest event in contributing to the AEP does not change with return period and that the measure of OEP/AEP can be very variable from year to year. The measure of OEP/AEP is not a good measure for assessing any potential changes in the relative importance of a single storm, and hence any changes in clustering, with an increasing return period of a seasons AEP. It should also be noted that as these results come from just one single climate model more robust conclusions could be made from applying our methods to a greater number of climate models.

It has been shown in several studies (Leckebusch et al., 2007; Pinto et al., 2012) that loss potentials associated with European windstorms would increase under future climate conditions. Based on low resolution ECHAM5 simulations, Karremann et al. (2014a) provided evidence that the combination of higher single losses and clustering in a warmer climate would lead to significantly shorter return periods for storm series affecting Europe. As in the present study we only focussed on wind-

15  storms under current climate conditions, it would be pertinent as a next step to evaluate if the tendency towards an increase in clustering holds true for the new high resolution CMIP6 climate projections.

*Competing interests.*  There are no competing interests at present.

*Acknowledgements.*  M.D.K.P is funded by NERC via the SCENARIO DTP (NE/L002566/1) and co-sponsored by Aon Benfield. J.G.P. thanks the AXA Research Fund for support. L.C.S is funded by the ERA4CS WINDSURFER project and the National Centre for Atmo-
20  spheric Science. We thank ECMWF for their ERA-Interim Reanalysis data (https://http://apps.ecmwf.int/datasets). All other data and code are available from the authors (m.d.k.priestley@pgr.reading.ac.uk).





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





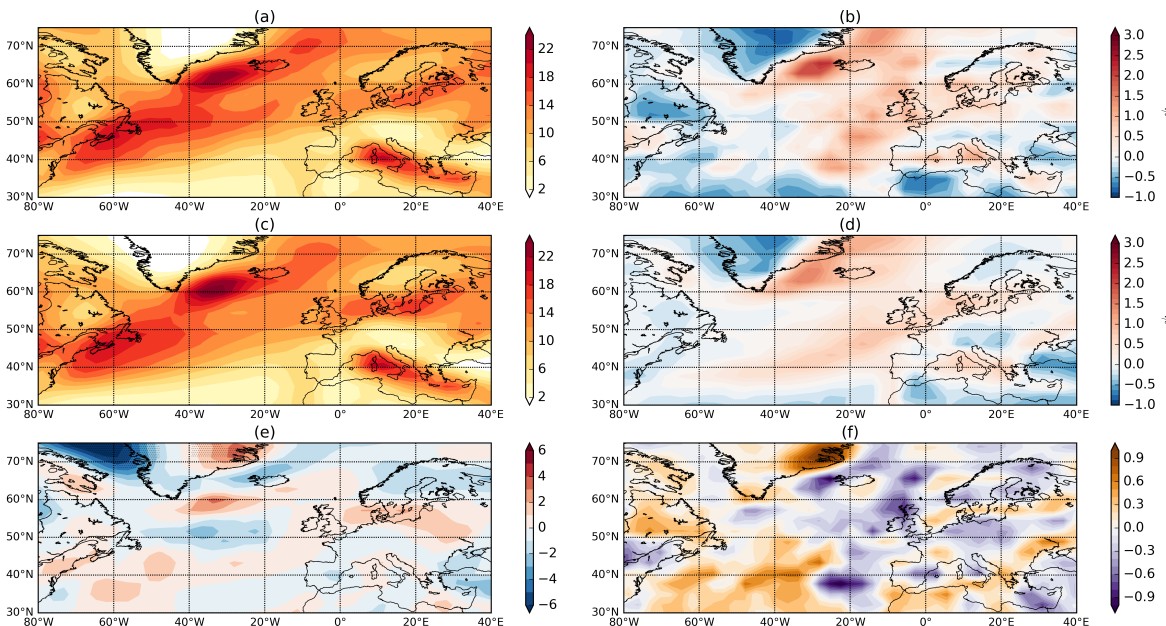

**Figure 1.** Track density (a) and associated dispersion (b) for ERA-Interim DJF storm track.(c), (d) same as (a), (b) but for HiGEM. (e) is the difference in track density (HiGEM-ERA-Interm), stippling indicates where the two datasets are different at the 95% level. (f) is the difference in dispersion (HiGEM-ERA-Interim). Units for (a), (c), and (e) are cyclones per month per 5° spherical cap.



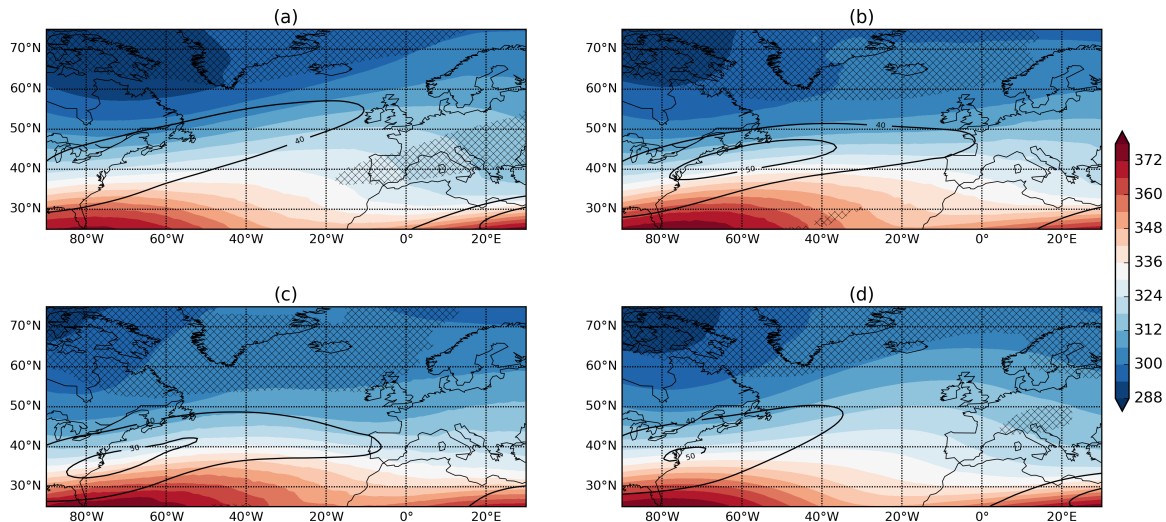

**Figure 2.** Dynamical composites of clustered days at (a) 65°N, (b) 55°N, and (c) 45°N. (d) Climatology. For all panels the colored contours are $\theta$ on the 2 PVU surface (K). Black contours are the 250 hPa wind speed, starting at 40 m s$^{-1}$ and increasing by 10 m s$^{-1}$. The crossed hatchings are where RWB was occurring on at least 30% of days.



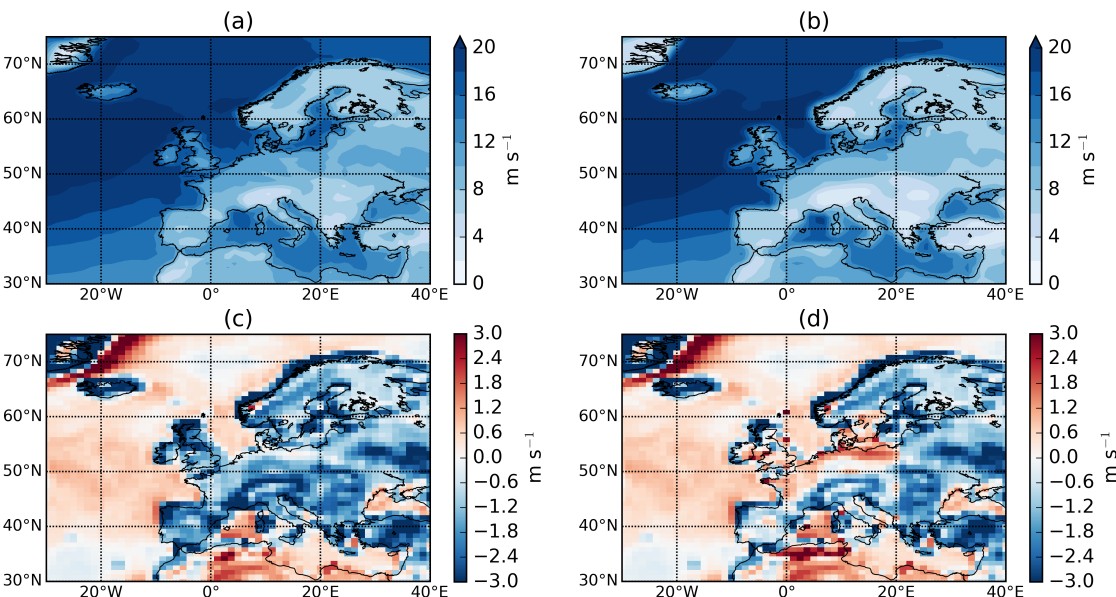

**Figure 3.** 98th percentile of 10-metre wind speed (m s$^{-1}$) for the DJF climatology of ERA-Interim (a) and HiGEM (b). (c) is HiGEM-ERA-Interim using the raw HiGEM wind speeds. (d) is HiGEM-ERA-Interim using the HiGEM winds that have been scaled by 18.75%.





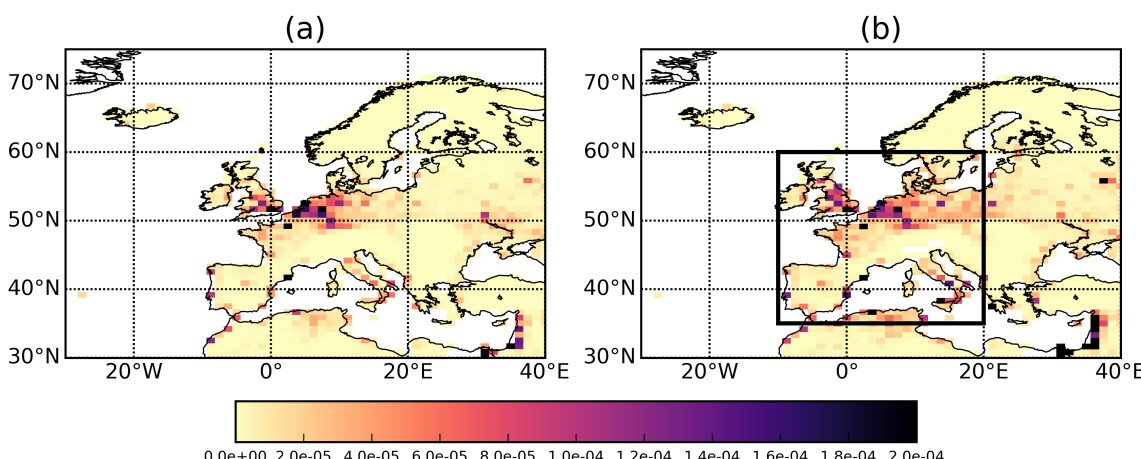

**Figure 4.** DJF average of 6-hourly SSI for ERA-Interim (a) and HiGEM (b). The black box region in (b) is our SSI calculation region.





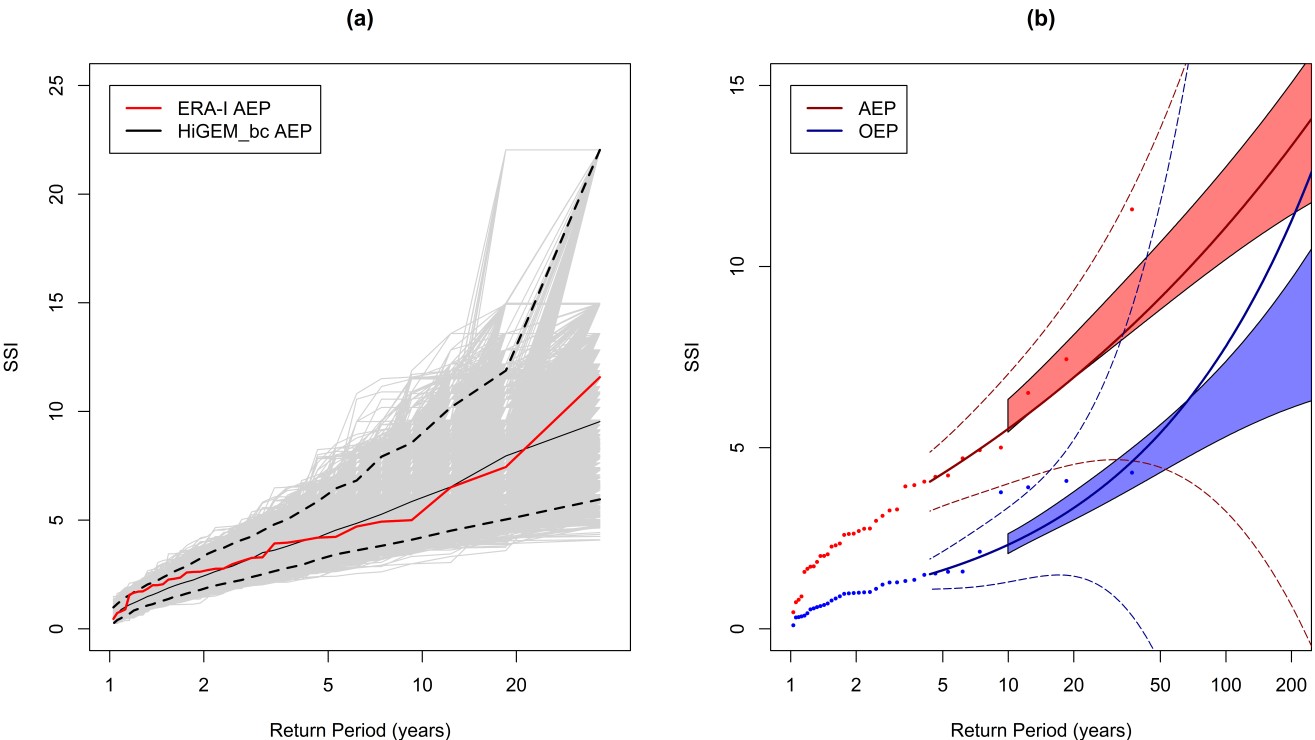

**Figure 5.** (a) Return periods of AEP for ERA-Interim (red line). The light grey lines are the 10,000 bootstrap samples of the HiGEM_bc AEP. The black dashed lines are the associated 95% confidence intervals of the HiGEM_bc AEP and the black solid line is the median. (b) Return periods of the AEP (red points) and OEP (blue points) of ERA-Interim. The solid red and blue lines are GPD fits applied to the AEP and OEP using an 70th percentile threshold. The dashed red and blue regions are the associated 95% confidence intervals. The shaded red and blue regions are the 95% confidence intervals of the HiGEM_bc AEP and OEP GPD fits. The OEP and AEP are sorted independently for both ERA-Interim and HiGEM. The GPD fits and confidence intervals are only plotted only above the GPD threshold of the 90th percentile.





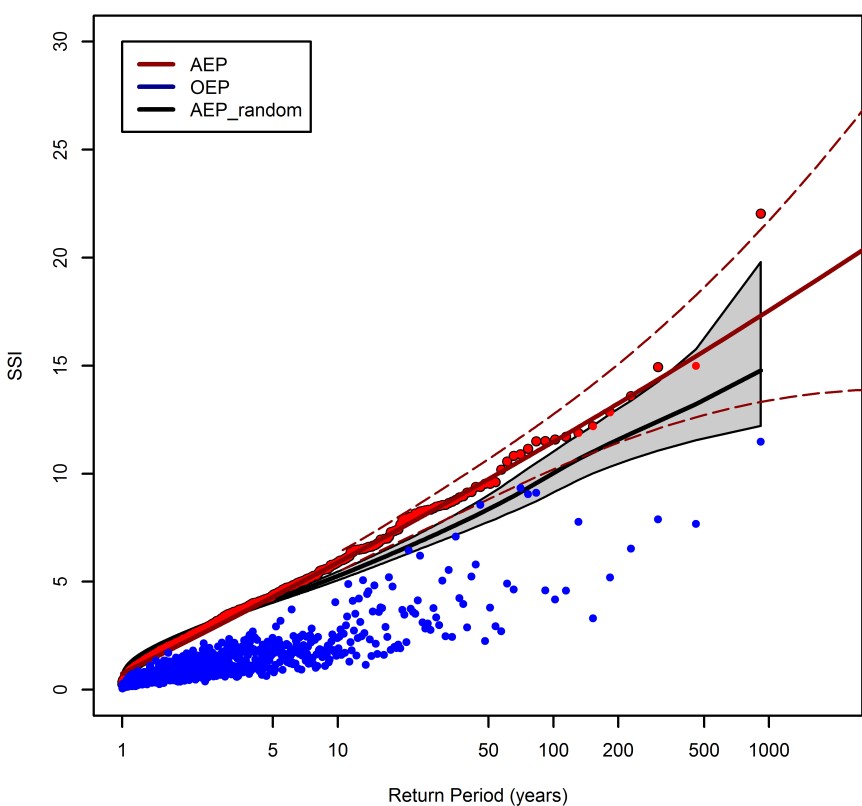

**Figure 6.** Return periods of the AEP (red points) and OEP (blue points) for HiGEM_bc. The OEP and AEP are sorted according to AEP magnitude. The solid red line is the GPD fit applied to the AEP using a 90th percentile threshold. The dashed red lines are the associated 95% confidence intervals. The black line represents the mean of 10,000 non-replacement random samples of the HiGEM_bc AEP data. The surrounding shaded grey region represents the 95% confidence interval of these 10,000 samples. The GPD fit and confidence intervals are only plotted above the GPD threshold.




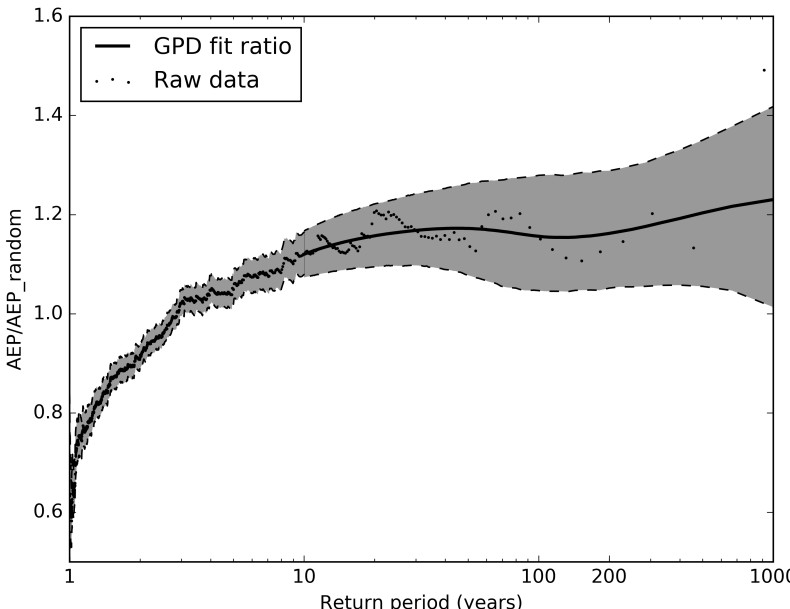

**Figure 7.** The ratio of the model AEP to the 10,000 random samples of the AEP for increasing return period. Black dots are the raw data points. The dark grey region below 10 year return period indicates the 95% confidence interval of the raw AEP/AEP_random. Above 10 year return period the dark grey shaded region bounded by the black dashed lines is the 95% confidence interval using the fitted AEP (red line in figure 6) in the calculation and the black solid line is the median of the spread. Confidence intervals from the GPD fits are only shown above the GPD threshold.





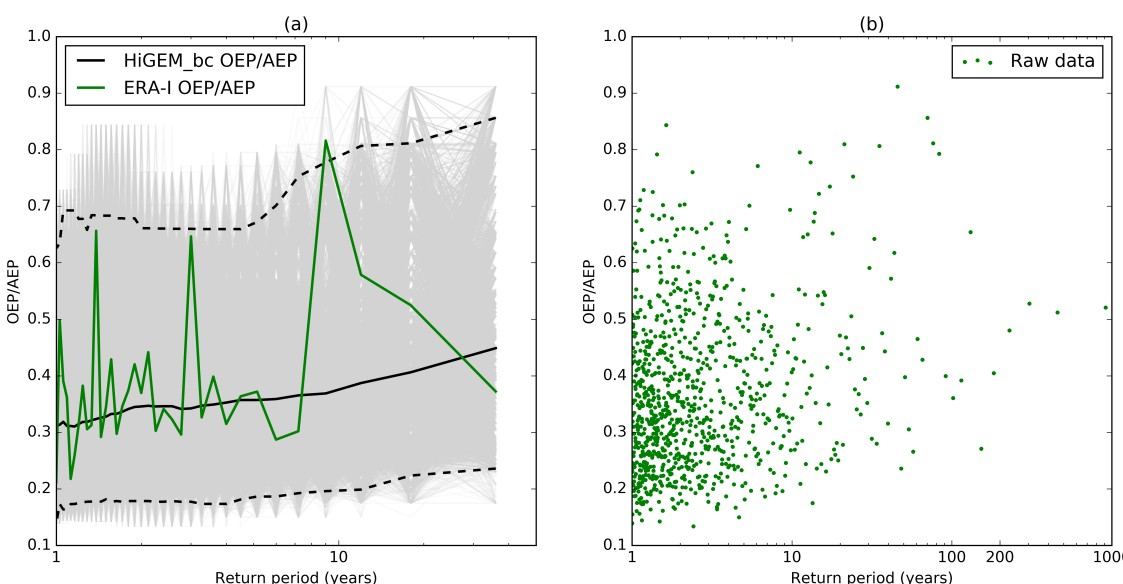

**Figure 8.** (a) Return periods of OEP/AEP for ERA-Interim (green line). The light grey lines are the 10,000 bootstrap samples of the HiGEM_bc OEP/AEP. The black dashed lines are the associated 95% confidence intervals of the HiGEM AEP and the black solid line is the median. (b) Return periods of the OEP/AEP ratio for HiGEM_bc (green points).





## Appendix A

### A1

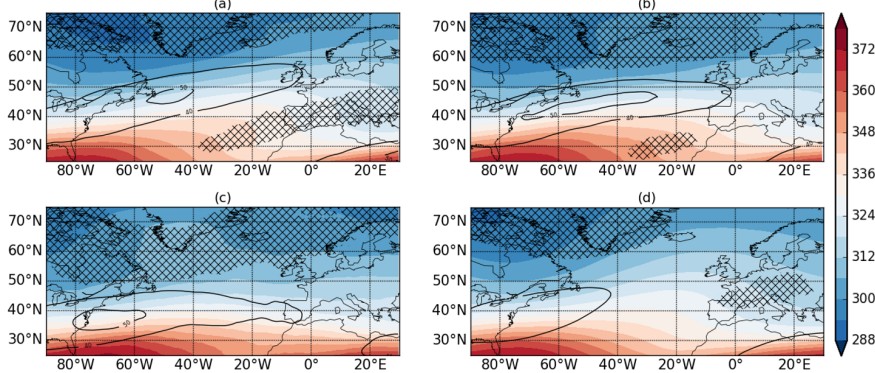

**Figure A1.** Composites of clustered days from ERA-Interim at (a) 65°N, (b) 55°N, (c) 45°N and (d) Climatology. Contours and hatchings are the same as figure 2.



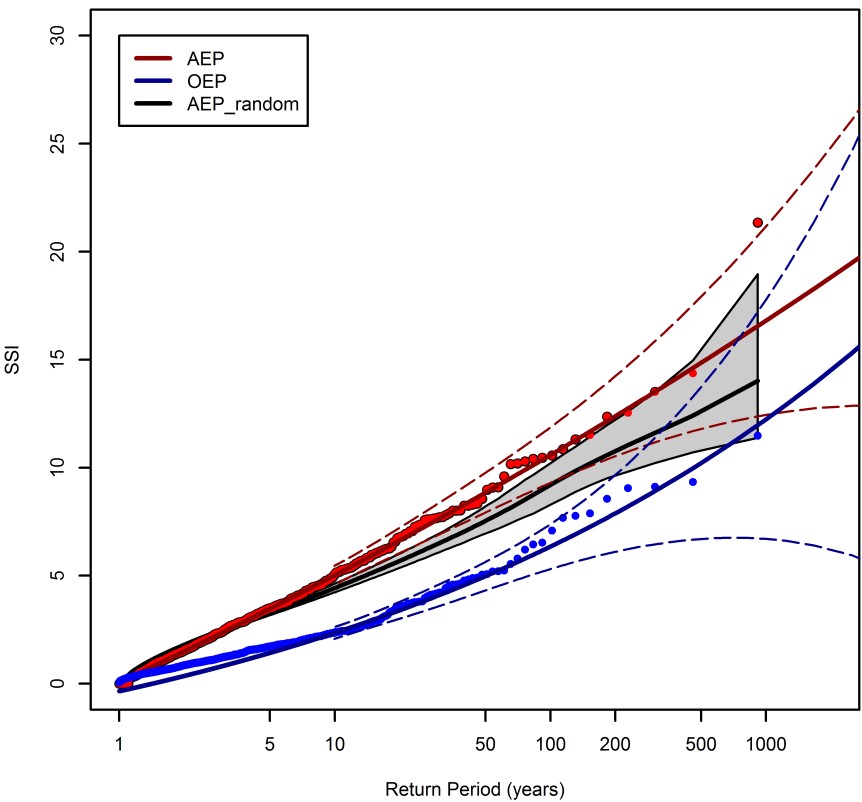

**Figure A2.** As figure 4, but only retaining the top three events of every year.