# Peer review of "The role of serial European windstorm clustering for extreme seasonal losses as determined from multi-centennial simulations of high resolution global climate model data"

_Natural Hazards and Earth System Sciences, 2018_

## Referee Comment (RC1) · Anonymous Referee #1 · 15 Jul 2018

This paper presents an analysis of temporal clustering of extratropical cyclones in the North Atlantic and the associated windstorm losses over central Europe. The studies shows the seasonally aggregated losses are substantially underestimated if temporal clustering is not taken into consideration. Also the relative contribution of the cyclone resulting in the highest losses per season to the overall seasonal losses is investigated. This contribution is very variable and ranges between 25 to 50%. The study makes use of decadal hindcasts to analyze hundreds of years of present day simulations and statistics based on this large sample are very robust. The quality of the text, the figures and the science is high. The only point that should be scrutinized is the GPD fit to the ERA-interim data. This analysis is not convincing and not necessary to support the

main points of the paper.

Minor points: Title: Suggest to add that the clustering refers to clustering in time P2L7 The most severe seasons in terms of the total windstorm loss P2L31 these events reference unclear P3L4/5 incomplete sentence P3l18 The question is incomplete contributes more than what? P5l25 Why is this threshold sensible? Why not set these grid-points to NaN? P6L8 Why a 72 hour threshold? P6 L36ff What is the base time for the analysis? Seasons, months? P7l16 How can this proportion become negative? Should it not be always positive? P8L15ff How do you decluster the extremes to use only independent values for the GPD estimate, especially as your time-series are clustered in time. See e.g. Ferro and Segers https://rss.onlinelibrary.wiley.com/doi/abs/10.1111/1467-9868.00401 P8L15 The GPD fit to ERA-interim seems a bit of a coup de main to me, especially considering that you are bias correcting the data beforehand thereby potentially introducing substantial uncertainty. Also the 70th percentile threshold for the fitting seems to be extremely low. Did you test this threshold? I would recommend fitting the GPD only to the model data, the results are convincing enough (even more convincing) without the this analysis. P8L27 well please quantify P9L19 Please define clustering for the readers not familiar with Priestley et al 2017 P9L30 slight please quantify P10L18 "this is balanced" be very careful, as soon as the exposure comes into play, the exact location matters a lot and you might introduce substantial artifacts. Are your results qualitatively independent from this bias correction? P11L24 increase between what and what? P11l33 marginally lower as would be expected . . . -> I do not understand this sentence P13l2 the 3 year P14l1 suggest: the importance of temporal clustering on seasonal timescales P14l28ff This sentence is unclear Figure 4: what are the units of SSI? What happens in the eastern Mediterranean? Figure 5a: Please add units

---

## Referee Comment (RC2) · Anonymous Referee #2 · 27 Aug 2018

Review of paper

"The role of European windstorm clustering for extreme seasonal losses as determined from a high resolution climate model"

by M. D. K. Priestley et al.

submitted to NHESS

This study uses a large set of HiGEM present-day climate simulations to estimate extreme seasonal losses due to windstorms and in particular due to the clustering of windstorms. As a main result, it is shown that the clustering of storms leads to peak

accumulated seasonal losses that are up to 20% larger than if the storms were randomly occurring. This result is interesting and the methodological approach to use almost 1000 years of simulated data with a fairly high-resolution model is sophisticated. Therefore I recommend the study for publication. I was just somehow disappointed about the quality of the writing. Many sentences are surprisingly fuzzy (surprising, because of the excellent team of co-authors). I therefore ask the authors to carefully revise their paper with the intention to explain things better to the hopefully large future readership. Some clarity issues are mentioned below.

Comments:

title and, e.g., p. 1 line 5: you call your model "high resolution". I understand why, but some colleagues think that high resolution is the km-scale, and in a few years this will be reality. I find the almost 1000 years that you have available for your evaluations more impressive than the HiGEM resolution of about 1 degree. Would it not be worth emphasizing this more in the title?

p. 1 line 3: "affect one area in a period of time" not sure that this is the best definition of serial clustering?

p. 1 line 7ff: this is a very long a complicated sentence. Please rephrase. How can a "loss-based metric" be "based on meteorological variables"? Then it is a metric for loss, but not loss-based?

p. 1 line 9: here it is completely unclear that the 918 years do not correspond to one long transient millennium simulation but rather to a large ensemble of present-day climate conditions. Please clarify.

p. 1 line 13: "return periods" of what?

p. 1 line 16: it is difficult to understand what you mean by "random realizations of the HiGEM data". This first sounded to me as if you were running HiGEM with randomly perturbed physics. Since this sentence, in my opinion, is the key result of the study, it

is important to explain this "random realization" much better in the abstract.

p. 1 line 18: usually you give values in %, so why here not 25-50%?

p. 2 line 3: US $?

p. 2 line 7: "space of time" –> "time interval"?

p. 2 line 26: "amounts of RWB" –> maybe "frequency of RWB"?

p. 3 line 4: word missing after "comprehensive"

p. 3 line 17: I am lost with this question. Do you mean "Does windstorm clustering contribute more to losses in seasons with large accumulated losses?"

p. 3 line 20: what is an "increasing return period winter season"?

p. 3: is it a good idea to mix 4 x 59 years of transient simulations with decadal hindcasts? How can you justify that this leads to a good statistical distribution of present-day climate variability? Wouldn't it be better to only use the decadal hindcasts as a more homogeneous dataset?

p. 3 line 28: I failed in understanding how you get in the end 918 years: 4 x 59 = 236 years are from the transient runs; how do the remaining 682 years distribute across the 4 ensemble members initialized between 1960 and 2006?

p. 5: there is unnecessary repetition between lines 9ff and 24ff.

p. 6 lines 5 and 6: why are these probabilities? it seems that these are losses.

p. 6 line 8: spelling of SSI

p. 6 line 14 and throughout the paper: most likely also in NHESS "Figure" is written with a capital F.

p. 6 line 27: unclear, "storm track numbers" in what time period?

p. 7 line 28: why "so"?

p. 8 line 17: "M. and G. (2009)"??

p. 8 line 23: not sure that predictability can have skill, maybe "skill in ... predictions"

p. 8 line 27: "contrast" –> maybe "agree"?

p. 9 lines 14-30: here the link to the rest of the paper is not immediately clear and there seems to be some repetition between the two paragraphs. I suggest to merge and shorten them and to explain the reader why this analysis of RWB is relevant for the main part of this study.

p. 10 line 10: how can 10-m winds be compared/contrasted with 925-hPa winds?

p. 10 line 20: here "HiGEM" is used as a symbol for wind speed (if I understand correctly), which is strange. Before you used $v\_{i,j}$ - the same could be used again. Instead of "_bc" I suggest to use e.g. ˆ*. And $p\_{98}$-overbar is not explained.

p. 10 line 24: west –> east?

p. 10 line 26: delete "how"o

p. 11: I am not an expert in statistics, but is it a good idea to calculate return periods of 50y from a 36-y dataset?

p. 11 line 24: I don't understand "the increase in the largest event is by ... 100%"

p. 11 line 33: what is "very marginally"?

p. 12 line 4: should read "clustering to the ..."

p. 12 line 5: to me the notation "AEP_random" looks a bit like computer code. Why not AEP with subscript r?

p. 12 line 29: spelling of "entire"

p. 13 line 1: "as a result" can be omitted

p. 14 line 17: not sure that I understand "scaling by 18.75%". I would understand

"scaling by a factor of 1.1875" or "uniform increase by 18.75%"

p. 14 line 31: to me, the 10-20% effect of clustering is surprisingly small. I find it an interesting result that this effect is not larger. Maybe you can discuss this a bit more. I then find the "strong implications" on p. 15 line 2 a bit exaggerated, since 10-20% might be below the general uncertainty level (for instance in the evolution of population density).

―――――――――――――――――――

---

## Author Comment (AC1) · 2 Oct 2018

Dear Reviewer,

We thank you for the comments and suggestions. Please find a full response to all the comments, and a marked version of the manuscript, in the supplement below.

Please also note the supplement to this comment:
https://www.nat-hazards-earth-syst-sci-discuss.net/nhess-2018-165/nhess-2018-165-AC1-supplement.zip

---

## Author Response (AR1)

**The role of serial European windstorm clustering for extreme seasonal losses as determined from multi-centennial simulations of high resolution global climate model data**

Matthew D. K. Priestley, Helen F. Dacre, Len C. Shaffrey, Kevin I. Hodges, Joaquim G. Pinto

**Response to reviewer 1**

Dear Reviewer,

We thank you for the comments and suggestions that you have made to our manuscript, which have helped improve its quality. Please find below a response to all of your comments and questions raised. Any page and line numbers refer to the initial NHESSD document. The italicised black text are the comments to the manuscript. Our responses are in red with any changes described. An amended version of the manuscript has also been uploaded to highlight the changes. In the marked version, text which has been removed has been struck through, with new additions being in red.

*This paper presents an analysis of temporal clustering of extratropical cyclones in the North Atlantic and the associated windstorm losses over central Europe. The studies shows the seasonally aggregated losses are substantially underestimated if temporal clustering is not taken into consideration. Also the relative contribution of the cyclone resulting in the highest losses per season to the overall seasonal losses is investigated. This contribution is very variable and ranges between 25 to 50%. The study makes use of decadal hindcasts to analyze hundreds of years of present day simulations and statistics based on this large sample are very robust. The quality of the text, the figures and the science is high. The only point that should be scrutinized is the GPD fit to the ERA-interim data. This analysis is not convincing and not necessary to support the main points of the paper.*

**Minor points:**
1. *Title: Suggest to add that the clustering refers to clustering in time*
We have added the word 'serial' and also re-worded the title in accordance with the comments from reviewer 2, comment 1. The title now reads 'The role of serial European windstorm clustering for extreme seasonal losses as determined from multi-centennial simulations of high resolution global climate model data'.

2. *P2L7 The most severe seasons in terms of the total windstorm loss*
The start of this sentence has been changed from 'The most severe seasons...' to 'The most severe seasons, in terms of total windstorm loss, ...'.

3. *P2L31 these events reference unclear*
'These events...' has been changed to 'Clustering...' in order to clarify what events are being referred to.

*4. P3L4/5 incomplete sentence*
The words 'spatial coverage' have been added. The sentence start now reads 'With this aim, the historical reanalysis datasets provide a comprehensive spatial coverage…'. This also addresses comment number 11 from reviewer 2.

*5. P3l18 The question is incomplete contributes more than what?*
The question has been rephrased for clarity. It now reads as 'Does windstorm clustering contribute more to losses in Europe for winter seasons with large accumulated losses?'. This is also a point raised by reviewer 2, comment 12.

*6. P5l25 Why is this threshold sensible? Why not set these grid-points to NaN?*
This threshold was chosen based on the results of Karremann (2015). They discuss how wind damage generally occurs for gusts speeds above 21 m/s. This gust speed relates to sustained wind speeds between 8 m/s and 11 m/s. Their analysis then concludes that 9 m/s to be the best minimum threshold for the SSI for all regions across Europe. These grid points are not set to NaN as we want to calculate an SSI values at all European grid-points, therefore setting the threshold to NaN would not allow us to do this.

*7. P6L8 Why a 72 hour threshold?*
The 72-hour threshold is commonly used by re-insurance companies to define the duration of an 'event' (Mitchell-Wallace et al., 2017). This is stated in the text.

*8. P6 L36ff What is the base time for the analysis? Seasons, months?*
The dispersion is calculated using the storm track density in units of storms per month. This value of storms per month is an average for each DJF winter season. The text has been clarified in the text to read 'This relates the variance ($\sigma^2$) in storm track density (average number of storms per month in a single DJF season) to the mean ($\mu$) storm track density'. This has also been changed to be in accordance with comment 20 from reviewer 2.

*9. P7l16 How can this proportion become negative? Should it not be always positive?*
Yes, this should always be positive. This should read that the distinction is in values above or below 1 for over/under dispersive behaviour of storms. This has been changed in the text.

*10. P8L15ff How do you decluster the extremes to use only independent values for the GPD estimate, especially as your time-series are clustered in time. See e.g. Ferro and Segers https://rss.onlinelibrary.wiley.com/doi/abs/10.1111/1467-9868.00401*
Independent values are used for the GPD estimate by the definition of the AEP and OEP only taking one value per winter season, therefore making them fully independent from the rest of the data. As the AEP and OEP provide one value per winter season, they are not related to the other seasons and hence all values are independent for the GPD estimations.

*11. P8L15 The GPD fit to ERA-interim seems a bit of a coup de main to me, especially considering that you are bias correcting the data beforehand thereby potentially introducing substantial uncertainty. Also the 70th percentile threshold for the fitting seems to be extremely low. Did you test this threshold? I would recommend fitting the GPD only to the model data, the results are convincing enough (even more convincing) without this analysis.*
The reason why we include this figure was to enable a comparison with the GCM data, and display the advantages of the longer dataset. Still, we agree with the reviewer that the GPD is not convincing and thus does not need to be shown and discussed so prominently. Subsequently, we have decided to split figure 5 up, retaining figure 5a in the main text and move

figure 5b to the supplementary material. This has been done for the purpose of continuity in the text with some minor changes made to reflect this. Some of the details regarding the visual details of figure 5b (now figure A2) have been removed from the text to improve the readability of this section. We decided against removing figure 5b altogether from the paper to keep consistency. Moreover, threshold for fitting the GPD to the ERA-Interim data was tested, and using a threshold consistent with that applied to HiGEM (90th percentile) did not result in a good fit due to the very small data sample. We have thus kept the threshold for figure A2 (old 5b)

*12. P8L27 well please quantify*
The wording has been changed slightly to now read 'agree…' as opposed to the original 'contrast well…' to remove the value judgement. In addition a quantification of the agreement has been added 'agree (anomalies within ± 2 cyclones per month)'. This is also in agreement with comment 24 from reviewer 2.

*13. P9L19 Please define clustering for the readers not familiar with Priestley et al 2017*
This is discussed in the introduction (paragraph 3) with a definition of clustering provided and the key findings of the Priestley et al. (2017) paper are given. A further quantification has been added to the start of this paragraph that this is '(as discussed in section 1)'.

*14. P9L30 slight please quantify*
This has been clarified in the text. It now reads 'albeit with a lower frequency of anticyclone RWB on the southern flank of the jet'.

*15. P10L18 "this is balanced" be very careful, as soon as the exposure comes into play, the exact location matters a lot and you might introduce substantial artefacts. Are your results qualitatively independent from this bias correction?*
We have slightly rephrased the sentence to remove ambiguity, It now reads 'the difference have changed sign and are now positive, there are also some regions that still have negative anomalies, resulting…'. Due to the bias correction being a uniform scaling to the wind data only, all the results examining the SSI are independent from the bias correction. Figure 1 shown at the end of the responses is a version of figure 7 in the main article with no scaling applied, and also no population weighting. You can see how the results are qualitatively very similar to those presented in the article with an average value of AEP/AEP_random of approximately 1.2 at a return period of 200 years.

*16. P11L24 increase between what and what?*
This is to illustrate the increase in the largest AEP and OEP from ERA-Interim to HiGEM. To improve clarity the text has been reworded to 'For example, the 918 year return period season in HiGEM is approximately twice the magnitude of the 1 in 36 year season in ERA-Interim.'. This change is also in accordance with comment 31 from reviewer 2.

*17. P11l33 marginally lower as would be expected… -> I do not understand this sentence*
As we are removing a large portion of the events (i.e. setting small events to have a magnitude of 0) that make up each season this results in a reduction of the AEP value compared to when including the full number of events. The sentence has been slightly changed to aid clarification to 'The magnitudes of the reduced event AEPs are marginally lower than the original AEPs, as would be expected…'.

*18. P13l2 the 3 year*
This has been changed in the text to 'above a return period of 3 years…'.

*19. P14l1 suggest: the importance of temporal clustering on seasonal timescales*
We have changed the sentence to 'The aim of this study is to investigate the importance of serial clustering on seasonal timescales for high…'.

*20. P14l28ff This sentence is unclear*
We have removed the sentence 'The absolute difference between the AEP and OEP is increasing with return period.' In order to improve the clarity and readability of this conclusion.

*21. Figure 4: what are the units of SSI? What happens in the eastern Mediterranean?*
There are no units to the SSI. (V/V$_{98}$) would be unitless. In addition the population density was normalised in order to reduce the magnitude of the final loss value. Therefore the SSI value is just a magnitude of event or seasonal loss which can be compared with the other events/seasons. With regards to the high SSI values in the eastern Mediterranean, we have found that these grid points have a very long tail in the 10-metre wind speeds compared to all of northern and northwestern Europe (7-8 m/s across E. Med., compared to 6-7m/s across NW Europe). This long tail means that high SSI values will be generated in this region. However, this is not a region for which the SSI is suitable (it is stated in the text that it is a 'loss proxy for European windstorms'), therefore these spurious values can be ignored and are not considered in our analysis.

*22. Figure 5a: Please add units*
Please see above comment regarding the units of the SSI.

[Figure]

Figure 1. As figure 7 in Priestley et al. (2018), but with no bias correction applied to the 10-metre wind speeds. No population density scaling has been applied either.

The 10-20% has large implications. If you consider that the average insured loss from European windstorms is ~€1.5 billion then the difference of 10% would be an underestimation of losses by €150 million. This is just the average for years within recent living memory, so for the larger loss seasons which are present in HiGEM, this would equate to a considerably larger value. Hence a misrepresentation by loss modellers would result in a vast underestimation of losses in a monetary sense. The monetary implications have also been included in the text. We have also removed the word 'strong' from 'strong implications'. With regards to the uncertainty of population density we have done additional analysis to show this result is actually rather insensitive to the population density scaling (see figure 1 below) and the same conclusions can be made. With this in mind, we do not

believe any differences in evolution of population density in recent years for central and western Europe over the last 30 years (for which the population density is available) would result in different results to those we have presented.

[Figure]

Figure 1. As figure 7 in Priestley et al. (2018), but with no population density scaling applied.

**References**

[revised manuscript text omitted]